# Hematological and CD$_4$+ T- cell count reference interval for pregnant women attending antenatal care at Hawassa University Comprehensive Specialized Hospital, Hawassa Southern Ethiopia

Yidnekachew Fiseha Gebere[1]☯, Lealem Gedefaw Bimerew[2‡], Wondimagegn Adissu Malko[2‡], Demissie Assegu Fenta[3]*

1 Department of Medical Laboratory, College of Medicine and Health Science, Hawassa University, Comprehensive Specialized Hospital, Hawassa, Sidama Region, Ethiopia, 2 School of Medical Laboratory Sciences, Institute of Health Faculty of Health, Jimma University, Jimma, Oromiya Region, Ethiopia, 3 School of Medical Laboratory Science, College of Medicine and Health Science, Hawassa University, Hawassa, Sidama Region, Ethiopia

☯ These authors contributed equally to this work.
‡ These authors also contributed equally to this work.
* demissieasegu@yahoo.com

**Data Availability Statement:** All relevant data are within the paper.

## Abstract

### Background

Pregnancy is a state characterized by physiological, hematological, and immunological changes. However, the reference intervals (RI) being used in clinical practice in Ethiopia are derived from non-local general populations. Therefore; this study was aimed to determine the reference interval of hematological and immunological profiles among healthy pregnant mothers attending Hawassa University Hospital.

### Methods

A cross-sectional study in a total of 360 healthy pregnant women was enrolled from January to April 2019, at Hawassa University hospital. Sociodemographic and obstetric data were collected using a structured questionnaire. Blood samples collected from each participant were used to define the hematological parameters. The median and 95% intervals were calculated for the immunological and hematological profiles. P-value 0.05 was considered statistically significant.

### Result

A total of 360 healthy pregnant women were enrolled in this study. The age range of the participants was 18–45 years. 342(95%) were married and 270 (75%) of the participants were multigravida. The overall median CD4+ T-cell and total WBC counts (cells/mm3) were 602 and 7.58 respectively. The overall median value for lymphocytes, neutrophils, monocytes, eosinophils, and basophil count was (cells/mm3) was 2.21, 6.74, .63, .53, and 0.09

**Funding:** This research has been supported by Jimma University by finance for sample collectors and reagent purchases, material support was obtained from Hawassa University College of medicine and health science and no other external fund was secured for this work. The funder has no role in the design of the study and collection, analysis, and interpretation of data and in writing the manuscript. No additional external funding was received for this study.

**Competing interests:** All authors declare that no competing interests exist.

respectively. Whereas the median RBC and platelet count was $4.48 \times 10^6$/μLand $212 \times 10^6$/μL. The median value of hematological profiles in the first, second, and third trimesters was TWBC ($10^3$/μL) (7.90, 8.30, 8.65), RBC ($10^6$/μL) (4.5, 4.6, 4.62), and PLT ($10^3$/μL) (210, 209,161) respectively. The CD4 T cell count median value was (600, 598, and 591) in the first, second, and third trimesters. Significant changes were observed in hematological and immunological parameters between trimesters ($P < 0.05$).

## Conclusion

Significant changes were observed in hematological and immunological parameters between trimesters ($P < 0.05$). Considerable differences were also seen between the values in this study and other studies from Ethiopia and other countries, indicated the need for the development of local reference intervals for pregnant women.

## Introduction

Regular assessment of hematological and immunological profiles is an essential and common medical practice for the evaluation of the state of health is not only for sick but also healthy subjects [1] and used to describe fluctuations of blood analyte concentrations in well-characterized groups of individuals such as pregnant women [2, 3]. The reference ranges could also be important tools as biomarkers to assess disease progression or response to therapy and in the assessment of adverse reactions to therapy [4]. These parameters may vary depending on age, gender, race, environmental and genetic background [4]. The profiles can also reflect the nutritional, immunological, and hemostatic status of pregnant women [5]. Likewise, it is an important predictor of pregnancy outcomes [6]. Physiologically, activation of the renin-angiotensin-aldosterone system during pregnancy increases extracellular fluid [7] and consequently, plasma volume [8], maternal erythropoiesis [9], neutrophil apoptosis [10], platelet activation, and clearance [11] are enhanced during pregnancy. Thus the hematocrit (HCT) value, platelet-crit (PCT), counts of red blood cells (RBC), white blood cells (WBC), and platelets (PLT) are expected to change according to the degree of plasma volume expansion [7, 8]. and the amount of blood formed elements being added or removed [10, 11] from the circulation.

Release of young RBC and activation of platelets affect the readings of some hematological indices like mean corpuscular volume (MCV), red cell distribution width (RDW), mean platelet volume (MPV), and platelet distribution width (PDW) [5, 11]. However, the exact pattern of trimester change of these hematological indices, and others, remained ill-defined. This fact encouraged researchers in the field to investigate for trimester-specific, reference range for hematological profile during normal pregnancy [12, 13]. Pregnancy is a state characterized by profound hematological changes, and regulated inflammation which is consistently marked by a rise in the total number of circulating leukocytes, primarily due to the increase of neutrophils and monocytes. These phagocytic cells can also induce inflammatory activation and strengthen resistance to extracellular infections, but are also associated with the etiology of autoimmune diseases [14].

Immunity in pregnancy is physiologically compromised and may affect the $CD_4+$ cell count, as lower $CD_4+$ cell count has been reported in pregnant compared with non-pregnant females [15]. On the other hand, total lymphocyte counts are consistently reduced during pregnancy, primarily due to fewer circulating cytotoxic lymphocytes capable of directly

recognizing and targeting fetal antigens [14]. On the contrary CD4 T cells, the proliferation of regulatory lymphocytes is increased [16]. Pregnancy also constitutes a time of characteristic challenges for the human immune system that it is becoming increasingly apparent for expression and regulation of immune components during gestation are unique, especially during early and mid-gestation when the development of organ systems is sensitive to alterations in the immune system [17].

Normal reference intervals (RIs) of hematological and immunological parameters of one population might be the cutoff point of clinical decision for other populations [18]. Any deviation from the normal reference range is indicative of several human diseases and therefore constitutes important parameters for diagnosis and patient monitoring particularly in this era of evidence-based medicine [19].

Most of the developing countries including Ethiopia used pre-established hematological and immunological reference values from developed countries. However, these, countries do not have the same physiological and environmental patterns as developed countries. Their health conditions and the distribution of healthy populations are different from that of developed countries which leads to misdiagnosis and mistaken treatment [20]. As a result, an expert committee in the US and Clinical and Laboratory Standards Institute (CLSI) recommended that the reference range should be established for each age, gender, ethnicity, climate, geographical location or environment, and population [20, 21]. Furthermore, numerous studies in African, Asian, and Western countries reported that reference ranges vary considerably in different populations, within-population, groups, geographical regions, climate, race, and dietary habits [22]. Likewise, the normal ranges of red blood cell counts (RBCs), hemoglobin (Hb) concentration, hematocrit (Hct), mean cell volume (MCV), total white cell count (WBC), platelet, and CD4+ T- cell counts are known to vary with age, sex, dietary patterns, ethnic origin, genetic factors, geographical regions, climate and environmental factors [22].

Few studies conducted in Ethiopia among apparently healthy individuals attending health institutions described that the Ethiopian population has a high hemoglobin (HB) and hematocrit (HCT) levels when compared to some African countries and Western population [23, 24]. On the other hand, they have lower CD4+ T-cell counts and CD4+/CD8+ T-cell ratios [24].

Hematological and immunological reference intervals among different groups of the Ethiopian population are poorly defined. Few studies conducted in Ethiopia among apparently healthy individuals attending health institutions indicated that the presence of significant differences with those of the western, Asian, and most African countries [25–28].

Thus, the establishment of a hematological and immunological reference range specifically for the Ethiopian population and particularly for pregnant women has specific importance for the interpretation of laboratory test results and provision of quality services in health care delivery. However, clinicians and researchers in different parts of Ethiopia are still using hematological reference ranges of western countries without any validation mechanism. Therefore, lack of own reference ranges may result in unappropriated management of patients and pregnant women and another decision making in research findings. Therefore, there is a top urgent need to have baseline data of reference ranges on hematological and immunological profiles for healthy pregnant women at the local community level to predict and/or improve pregnancy outcomes during antenatal care. This study was aimed to establish the immunological and hematological reference intervals of apparently healthy pregnant women at Hawassa University hospital in Sidama region, Southern, Ethiopia.

## Materials and methods

### Study setting and population

A cross-sectional study was conducted from January to April 2019 at Hawassa University comprehensive specialized Hospital. The hospital is located in Hawassa city which is 275 km south of Addis Ababa, the capital city of Ethiopia. Geographically the City lays between 703' latitude North and 380 28' longitudes east and it is situated at an elevation of 1697 meters above sea level. Geographically the City lays between 703' latitude North and 380 28' longitudes east and it is situated at an elevation of 1697 meters above sea level. Based on the world bank group SFD report 2016, Hawassa has a population of 351,469, of whom 180,785 are women and 170,684 men [29]. Hawassa University comprehensive specialized hospital is the only referral hospital in the region with more than 500 beds in southern Ethiopia serving a population of about 18 million in the nearby regions of Oromiya and Somalia. The hospital has organized a special ANC clinic called MICHU which serves an average of 25 pregnant mothers daily from Hawassa town and its surroundings.

### Sample size determination and sampling technique

The sample size was determined according to the National Committee for Clinical Laboratory Standards (NCCLS), International Federation of Clinical Chemistry (IFCC), and Clinical Laboratory Standards Institute (CLSI) recommendations, with well-defined exclusion and partitioning criteria before the selection of the reference individual. Based on this guideline a minimum size of 120 participants was used for this study [30] and the study participants were partitioned into three subgroups based on their gestational age (trimesters) which includes 120 healthy pregnant mothers in each group to reach a total of 360 sample size. These participants were selected based on medical examinations conducted by the physician assigned in place and blood sample was collected from each participant and sent to the laboratory for the screening of infectious diseases such as HIV, HBV, HCV, malaria, and syphilis by trained laboratory technologist. A total of 360 participants who met the inclusion criteria and consented were recruited by convenience sampling technique. Of them 240 Pregnant women with any of the following conditions were excluded from the study due to current health status, blood pressure, diabetics, treatment (medication), working with hazardous chemicals, alcohol intake, presence of inherited health disorder in the family, tuberculosis, lymphadenopathy, weight loss, regular exercise, tobacco smoking, allergy manifestation, fever and infectious diseases such as (malaria, HIV, hepatitis B virus, hepatitis C virus, syphilis) and others who have histories of chronic illness.

### Data collection

Demographic data, information on drug history, and other health-related data were collected directly from the recruited participants using a structured questionnaire. Each participant has undergone thorough a medical interview and clinical examination by a physician. Variables such as age, educational status, occupation, gestational age, parity, food intake were collected. The blood pressure was measured using an automatic digital blood pressure monitor.

### Laboratory methods

Full blood count was performed using a KN-21N Hematology Analyzer (Sysmex, Kobe, Japan), a three-part auto analyzer able to test 19 parameters per sample including Hb concentration, PCV, RBC concentration, MCH, MCV, MCHC, WBC count, and PLT count.

Standardization, calibration of the instrument, and processing of the samples were done according to the manufacturer's instructions.

After written informed consent was taken from study subjects, approximately 5 mL of venous blood was collected from each study participant in the morning from 8:00 am to 11:00 am using $K_2$Ethylene Diamine Tetra acetic Acid ($K_2$EDTA) test tube for immunological and hematological determination and plane test tube for serological testing using a sterile syringe for a screening of HIV, HBV, HCV and Syphilis and processed within 3 hours of collection. The study participants were also tested for hemoparasites and intestinal parasites. Participants who become positive for one or combinations of the infectious disease were contacted by the physician who was working in a place for further treatment and management.

Full blood count was performed using Cell-Dyn Ruby (Abbott Laboratories, Chicago, IL, USA) a five parameter automated hematology analyzer which was standardized against a 4C® Plus blood control Including differential count (Neutrophils, Lymphocyte, Eosinophil, Monocyte, and Basophil), hemoglobin (Hb), hematocrit (Hct), and RBC indices (MCV, MCH, and MCHC) were determined. The immunological parameters were analyzed using the BD FACS presto cartridge; the $CD_4$+ cell cartridge contains dried fluorochrome-conjugated antibody reagents (Becton Dickinson, San Jose, CA, USA). The performance of the instrument was controlled by running control samples before the start of the test and the software identifies the cell population of interest and calculates $CD_4$+ cell absolute counts, $CD_4$+ cell percentage of lymphocytes, and hemoglobin concentration.

HIV infection was detected using rapid HIV-1/2 test kits as per the national algorithm for HIV testing. Briefly, whole-blood samples were screened for HIV using STAT-PAK™, ABON™ HIV1/2, and SD-Bioline HIV1/2 test kits following the manufacturer's instructions.

**HBsAg.** Serum HBV was screened using Wondfo rapid one-step HBsAg plasma test kit (Guangzhou Wondfo Biotech Co., Ltd, China) and HCV was screened by Wondfo one step HCV test strip (Guangzhou Wondfo Biotech Co., Ltd, China) for the qualitative detection of antibody to HCV in human plasma/serum. Syphilis was screened using the syphilis rapid test strip (serum/plasma) to detect IgM and IgG antibodies to *Treponema Pallidum* through visual interpretation of color development on the strip.

**Hemoparsites were checked by using** a drop of a blood sample and preparing thick and thin smears making the dried labeled, fixed, and stained with Giemsa staining solution, and examining using oil immersion (100x objectives) of the light microscope. But fresh stool samples were used for parasitological examination of intestinal parasites using the direct wet mount method.

## Quality control

To ensure the quality of data, training was given to data collectors before data collection structured pretested questionnaire was used for data collection. We used a standard operating procedure (SOP) for pre-analytical, analytical, and post-analytical procedures implemented during hematological and immunological test measurements. All samples were analyzed in one laboratory with similar analyzers and the same trained professionals. For these analyzers, daily initialization background check, three levels (tri-level) of commercially available whole blood quality control material (high, normal, and low) were used to check the analytical capability of the machine daily on startup.

## Data processing and analysis

Data from the questionnaires and laboratory results were checked for completeness and entered into Epidata 4.1 and exported to SPSS version 20 software (IBM Corporation,

Armonk, NY, USA) for analysis. A descriptive analysis was used to summarize the sociodemographic and clinical characteristics of the study participants. Reference intervals were computed using the nonparametric method. The mean, median, and standard deviation (SD) values were calculated for each immunological and hematological parameter. The 95th percentile reference intervals were determined. Analysis of variance (ANOVA) was used for any statistically significant difference in these parameters concerning the age and trimesters of the study participants. P-values, 0.05 were considered statistically significant for all variables.

### Ethics approval and consent to participate

Ethical approval for this study was obtained from Jimma University Institute of Health Ethical Review Board with a protocol number of IHRPGD/573/12/2018. A support letter was written from Jimma University to Hawassa University College of Medicine and Health Science comprehensive specialized hospital (HUCSH). A permission letter was obtained from the clinical and academic director of HUCSH to conduct the study. Informed written consent was obtained from each voluntary participant before enrolment with strict privacy. The data was handled with the assurance of confidentiality.

## Results

### Demographic and clinical characteristics

A total of 883 (334 children, 289 adults, and 260 geriatrics) study participants were included in the final statistical analysis for hematological RI estimation. From these, 430 were males and 453 were females. The mean age of the study participants was $27.61 \pm 18.5$ years (male = $28.5 \pm 18.9$ and female = $26.7 \pm 18$), with a range of 5–71 years

A total of 360 healthy pregnant women who fulfilled the inclusion criteria were enrolled in the final statistical analysis for hematological and immunological reference interval (RI) estimation. The age range of the participants in this study was 18–45 years old. Of these 116 (32.2%) were younger than 24 years, 152 (42.2%) were between 25 and 29 years, 68 (18.8%) were between 30 and 34 years, while 20(5.6%) were between 35 and 39 years and 4(1.1%) were between 40 and 45 years. From the total participants included in this study, 342(95%) were married and 18(5%) were unmarried, and 270 (75%) of them were multigravida and 90(25%) were Primigravida. All of the participants described have no smoking and alcohol consumption habits. According to the stage of pregnancy, 120 apparently healthy pregnant women were included from the first, second, and third trimesters respectively at the time of the study (**Table 1**).

### Hematological and immunological parameters

A total of 360 pregnant women participated in this study. The median and 95% RI ((2.5th–97.5th percentile) were established, for total WBC 6.1–10.9 x$10^3$/μL; platelet: 165–260 x$10^3$/μL; RBC: 3.43–5.53 x$10^6$/μL counts and for hemoglobin: (Hb) 11.5–13.4g/dl and hematocrit: 38.5–40.1% concentration WBC differential count of Neutrophil:6.54–7.26 x$10^3$/ μL, Lymphocyte: 2.21–2.91 x$10^3$/ μL and Eosinophil 0.53–0.63 x$10^3$/ μL. Immunological reference values were established for only one parameter and presented using median and 2.5th-97.5th percentiles where the median CD4+ T-cell count was 547–657 cells/mm$^3$. Furthermore; the total median value for total WBC count and Differential WBC count lymphocyte, Neutrophil, Monocyte, Eosinophil, and Basophil was 7.58 ×$10^3$/μL, 2.21×$10^3$/μL, 6.54×$10^3$/μL, 0.63×$10^3$/μL, 0.53×$10^3$/ μL, and 0.09×$10^3$/μL respectively. Whereas the median RBC and platelet counts were 4.48×$10^6$/μL and 212×$10^3$/ μL (**Table 2**).

**Table 1. Sociodemographic and behavioral characteristics of pregnant women at the ANC clinic of Hawassa University Hospital, Southern Ethiopia, 2019.**

| Parameters | Categories | Frequency(n) | Percent (%) |
|---|---|---|---|
| Age group | 18–24 | 116 | 32.2 |
| | 25–29 | 152 | 42.2 |
| | 30–34 | 68 | 18.8 |
| | 35–39 | 20 | 5.6 |
| | 40–45 | 4 | 1.1 |
| Gravida | Primigravida | 90 | 25 |
| | Multigravida | 270 | 75 |
| Parity | One child | 120 | 33.3 |
| | 2–3 children | 128 | 35.5 |
| | 4–5 children | 32 | 8.8 |
| | Others | 80 | 22.4 |
| Occupational status | Merchant | 40 | 11.6 |
| | Student | 48 | 13.3 |
| | Housewife | 168 | 46.4 |
| | Daily laborer | 22 | 6.1 |
| | Employee | 62 | 17.7 |
| | Others | 20 | 5.5 |
| Alcohol intake | Yes | 0 | 0 |
| | No | 360 | 100 |
| Cigarette smoking | Yes | 0 | 0 |
| | No | 360 | 100 |
| Gestational Age | First trimester | 120 | 33.3 |
| | Second Trimester | 120 | 33.3 |
| | Third Trimester | 120 | 33.3 |

**Trimester-based reference values for hematological and immunological parameters.**
According to the stage of pregnancy median and 95% reference range (2.5th–95th percentiles) for Total WBC were (7.90, 8.30, 8.65) × (103/μL), The median RBC count for each group was (4.5, 4.6, 4.62) × (106/μL), and for platelet count for each group was (210, 209,161) × (103/μL) in the first, second, and third trimester respectively.

The median hemoglobin and hematocrit reference values in each group of pregnancy were (11.9 gm/dL, 10.9 gm/dL, and 11.4gm/dl) and (36.4%, 33.1%, and 35.2%) for the first, second, and third trimester respectively. The MCV for the first, second, and third-trimester group was 75.9, 76.9, and 78.8fL, and for the MCH value for each group was 26.3, 27.1, and 27.3pg respectively. The median immunological reference values for CD4 T cell count was (600, 598, and 591× Cells/μL) in the first, second, and third trimesters. The reference value for hematological and immunological parameters using 95% CI as the 2.5 and 97.5 percentile was for total WBC $(6–11×10^3/μL, 5.9–11.3×10^3/μL, \text{ and } 6.1–12×10^3/μL)$, RBC $(3.5–5.5× 10^6/μL, 3.9–5.6 × 10^6/μL,$ and $3.4–5.7× 10^6/μL)$, Platelet $(162–258 × 10^3/μL, 161–257× (10^3/μL \text{ and } 165–261× (10^3/μL)$ in the first, second and third trimester respectively. While the median reference interval values of CD4 T-cell count in the first, second, and third trimester were (545–655 ×Cells/μL, 540–653×Cells/μL, and 536–646× Cells/μL) (**Table 3**).

## Intertrimester comparison of hematological and CD$_4$+ parameters

The median value and 2.5$^{th}$ and 97.5$^{th}$ percentile confidence interval of hematological and immunological parameters in the studied pregnant women were compared between each

**Table 2. The overall median and reference intervals of hematological and Immunological parameters among pregnant women at HUCSH, Ethiopia; 2019.**

| Parameters | Overall Reference Interval | | |
|---|---|---|---|
| | **Median** | **2.5** | **97.5** |
| WBCs($10^3$/µL) | 7.58 | 6.1 | 10.9 |
| Neutrophil ($10^3$/µL) | 6.54 | 6.01 | 7.26 |
| Lymphocyte ($10^3$/µL) | 2.21 | 1.53 | 2.91 |
| Monocytes ($10^3$/µL) | .63 | .43 | .91 |
| Eosinophil ($10^3$/µL | 0.53 | .49 | .63 |
| Basophil ($10^3$/µL) | 0.09 | 0.065 | 1.02 |
| Red blood cell ($10^6$/µL) | 4.48 | 3.43 | 5.53 |
| Hemoglobin (g/dl) | 12.0 | 11.5 | 13.4 |
| Hematocrit (%) | 35.8 | 30.6 | 40.1 |
| Mean corpuscular value (fL) | 77.2 | 67.4 | 87.5 |
| Mean cell hemoglobin (pg) | 26.9 | 24.1 | 30.5 |
| MCHC (g/dl) | 30.0 | 27.1 | 33.2 |
| Red cell distribution width (%) | 11.0 | 9.0 | 12.3 |
| Platelet Count ($10^3$/ µL) | 212 | 165 | 260 |
| Mean platelet value(fL) | 9.0 | 7.0 | 11.5 |
| $CD_4$+(Cells/µL) | 602 | 547 | 657 |

CD: Cluster of differentiation; MCHC: Mean Cell Hemoglobin Concentration

trimester. The differences between the median values across the respective three trimesters (first, second, and third) were 7.90 ×$10^3$/µL, 8.3×$10^3$/µL, and 8.65×$10^3$/µL for total WBC, respectively. Which was statistically significant between the first and second trimester (P = 0.008), first and the third (P-009), and the second and third trimester.

**Table 3. Trimester median values of immunological and hematological parameters among pregnant women at HUCSH, Ethiopia.** 2019.

| Parameters | | First trimester | | | 2nd trimester | | | 3rd trimester | | |
|---|---|---|---|---|---|---|---|---|---|---|
| | **Mean±SD** | **Median** | **95% CI 97.5** | | **Median** | **95% CI 97.5** | | **Median** | **95% CI 97.5** | |
| WBCs($10^3$/µL) | 8.8±1.56 | 7.90 | 6.0 | 11.0 | 8.3 | 5.9 | 11.3 | 8.65 | 6.1 | 12.5 |
| Neutrophil ($10^3$/µL) | 5.6±1.57 | 6.58 | 5.30 | 7.50 | 6.81 | 5.50 | 7.31 | 6.90 | 5.65 | 7.55 |
| Lymphocytes ($10^3$/µL) | 2.4±1.54 | 2.31 | 1.81 | 2.83 | 2.13 | 1.79 | 2.65 | 2.51 | 1.83 | 2.71 |
| Monocytes ($10^3$/µL) | 0.3±1.5 | 0.60 | 0.41 | 0.83 | 0.72 | 0.52 | 0.92 | 0.70 | 0.62 | 0.75 |
| Eosinophils ($10^3$/µL) | 4±1.6 | 5.0 | 3.5 | 6.1 | 5.1 | 3.9 | 6.3 | 5.5 | 4.5 | 6.2 |
| Basophil ($10^3$/µL) | 0.05±0.25 | 0.082 | 0.061 | 0.12 | 0.07 | 0.05 | 0.11 | 0.85 | 0.60 | 1.85 |
| Red blood cell ($10^6$/µL) | 4.6±0.6 | 4.5 | 3.5 | 5.5 | 4.60 | 3.9 | 5.6 | 4.62 | 3.4 | 5.72 |
| Hemoglobin (g/dl) | 12±1.5 | 11.9 | 10.9 | 13.0 | 10.9 | 9.9 | 12.0 | 11.4 | 10.4 | 12.6 |
| Hematocrit (%) | 36±4 | 36.4 | 30.1 | 45.3 | 33.1 | 28.1 | 39.3 | 35.2 | 30.5 | 40.3 |
| Mean corpuscular value (fL) | 84±5 | 75.9 | 65.9 | 85.9 | 76.9 | 66.9 | 86.9 | 78.8 | 68.3 | 88.8 |
| Mean Hgb concentration (pg) | 24±4 | 26.3 | 23.3 | 29.3 | 27.1 | 24.1 | 30.1 | 27.3 | 24.3 | 30.3 |
| MCHC (g/dl) | 31±2.8 | 30.9 | 30.3 | 33.9 | 29.9 | 26.5 | 32.5 | 30.3 | 27.3 | 33.3 |
| Red cell distribution width (%) | 11±1.5 | 10.9 | 9.0 | 12.9 | 11.1 | 9.1 | 12.1 | 11.5 | 9.5 | 12.5 |
| Platelet ($10^3$/ µL) | 217±45 | 210 | 162 | 258 | 209 | 161 | 257 | 213 | 165 | 261 |
| Mean platelet value (fL) | 9.2±5 | 9.5 | 7.5 | 11.5 | 10.3 | 8.3 | 12.3 | 10.5 | 8.5 | 12.5 |
| $CD_4$+(Cells/µL) | 614±92 | 600 | 545 | 655 | 598 | 540 | 653 | 591 | 536 | 646 |

CD: Cluster Differentiation; Hgb: Hemoglobin; MCHC: Mean Cell Hemoglobin Concentration; SD: Standard deviation.

The median RBC values across the three trimesters in the current study were $4.50 \times 10^6/\mu L$, $4.60 \times 10^6/\mu L$, and $4.62 \times 10^6/\mu L$, respectively, and there was statistically significant between the first and second (P = .009) and first and third trimesters (0.003), but not between second and third trimesters (P = .998). Whereas the median Hb value in the first and third trimester was 11.9 and 11.4 g/l respectively which were higher than the second trimester 10.9 g/l. It was statistically significant between the first and second trimesters (P = 0.004) and first and third trimesters (0.001*). The median PLT values in the first, second, and third trimesters were 210 $\times 10^3/\mu L$, $209 \times 10^3/\mu L$, and $213 \times 10^3/\mu L$ respectively. There was a statistically significant between the first and third trimester (P = 0.012) and the second and third trimester (P = 0.013), but not between the first and third trimester (P = 0.381).

The median CD4 T cell value in the respective trimesters of first, second, and third trimesters were 600 Cells/$\mu L$, 598 Cells/$\mu L$, and 591 Cells/$\mu L$ respectively. There was a statistically significant between the first and third (P = .001*) and the second and the third trimesters (0.003), but not between the first and the second trimesters (P = .0118) (Table 4).

## Discussion

Determining the reference intervals of hematological and $CD_4$+ T cell parameters have an indispensable role in the assessment of the health status of pregnant women and an individual, evaluation of disease prognosis, therapeutic drug monitoring, and selecting appropriate study participants in clinical trials for a certain population. Despite the fact; more than 80% of medical decisions rely on information provided by laboratory test results that are interpreted based on reference interval values, however; a scarcity of reference interval studies and the dependence on pre-established hematological and immunological reference ranges from other developed countries for individual group populations such as pregnant women in Ethiopia [31] is inappropriate, thereby leading to misdiagnoses resulting in wrong treatment.

This study, therefore, established the hematological reference values in apparently healthy pregnant women at Hawassa University College of medicine and health science comprehensive specialized hospital, Ethiopia.

The overall median value of RBC was (Table 2) lower than the study conducted in Addis Ababa ($4.58 \times 10^6/\mu L$) [32]. But; higher than the other studies conducted in Gondar ($4.37 \times 10^6/\mu L$) [33], India ($4.09 \times 10^6/\mu L$) [13], Libya ($4.11 \times 10^6/\mu L$) [34], and Nigeria ($3.5 \times \times 10^6/\mu L$). These discrepancies might be due to geographical, educational, economical status, nutrition,

**Table 4. Kruskal Walis test comparing reference interval for hematological and CD4 values between trimesters of pregnancy, at HUCSH, Ethiopia. 2019.**

| Parameters | Trimesters | | | P-Value | | |
|---|---|---|---|---|---|---|
| | Trimester one ($T_1$) | Trimester two ($T_2$) | Third Trimester ($T_3$) | $T_1$ Versue $T_2$ | $T_1$ Versues $T_3$ | $T_2$ Versues $T_3$ |
| WBC $\times(10^3/mm^3)$ | 7.90 | 8.3 | 8.65 | 0.008 | 0.009 | 0.002 |
| Neutrophil($10^3/mm^3$) | 6.58 | 6.81 | 6.90 | 0.002 | 0.004 | 0.013 |
| Lymphocyte ($10^3/mm^3$) | 2.31 | 2.13 | 2.51 | 0.006 | 0.012 | 0.015 |
| Red blood cells ($x10^6/mm^3$) | 4.5 | 4.60 | 4.62 | 0.009 | 0.003 | 0.998 |
| Hemoglobin (g/dl) | 11.9 | 10.9 | 11.4 | 0.004 | 0.000 | 0.957 |
| Hematocrit (%) | 36.4 | 33.1 | 35.2 | 0.000 | 0.005 | 0.000 |
| Mean corpuscular value (fl) | 75.9 | 76.9 | 78.8 | 0.179 | 0.000 | 0.142 |
| MCHC (g/dl) | 26.3 | 27.1 | 27.3 | 0.125 | 0.065 | 0.954 |
| Platelet ($x10^6/mm^3$) | 210 | 209 | 213 | 0.381 | 0.013 | 0.012 |
| CD4 T Cell count/$mm^3$ | 600 | 598 | 591 | 0.118 | 0.000 | 0.003 |

CD: Cluster Differentiation; MCHC: Mean Cell Hemoglobin Concentration; SD: Standard deviation, WBC: White Blood Cells

genetic difference, health care facility, and behavioral differences among study areas. Furthermore; the median RBCs count was slightly increased as the stage of pregnancy advances from the first trimester ($4.5\times10^6/\mu L$), second ($4.60\times10^6/\mu L$), and third trimester ($4.62\times10^6/\mu L$). This finding was in line with the study conducted in Addis Ababa [32], where the median value of RBC ($4.61\times10^6/\mu L$) in the first, ($4.86\times10^6/\mu L$) second, and ($4.46\times10^6/\mu L$) third trimester. Other similar findings were also reported from other studies conducted in Gondar [33], India [13], and Jamaica [35].

In our study, the change in the median hemoglobin value was not consistently increased or decreased across the trimesters of the first (11.9g/dl), second (10.9g/dl), and third trimester (11.4g/dl). A similar finding was reported from Addis Ababa [32], with a median value of 13.6g/dl, 12.6g/dl, and 12.97g/dl in the first, second, and third trimesters respectively. Whereas the overall median Hb value in the current study was slightly higher than a study conducted in India 10.3g/dl [13] and Libya 10.91g/dl [34]. But lower than the findings reported from Addis Ababa Ethiopia 16.4g/dl [36]. This difference could be due to the difference in altitude, nutritional status, genetic differences, and.analytical methods used for measurement.

The median MCV and MCH values increase as pregnancy advances from first to third trimesters in this study. This was comparable with the finding reported from Nigeria [37], and Sudan [38]. Inconsistent findings were reported from other Nigerian studies [39], and Ethiopian studies [36]. However; the total median MCV value in the current study was 77.2fL which was comparable with the study from Nigeria (78.3fL) [37] but; lower than the findings from Gondar (93.3fL) [33], Addis Ababa (90.6fL) [32], Libya (95.6fL) [34], and Sudan (82.7fL) [40].

The increased level of MCV and MCH with gestational age might be associated with the low prevalence of anemia in pregnant women and that of well-controlled supply of micronutrients like iron for the normal maintenance of hematologic profiles of the majority of the study pregnant women of this study relative to the dilution effect of their plasma volume.

The total median MCH value was (26.9pg) which was comparable with a study from India (26.92pg) [13] but; lower than the study reported from Addis Ababa (29.32pg) [32] and Gondar (30.2pg) [33].The median MCHC value (30g/dl) was considerably lower than that of their Libyan (36.2g/dl) [34], Indian (32,06g/dl) [13] Addis Ababa (32.33g/dl) [32], and Gondar (32.63 g/dl) [33] counterparts (**Table 2**).

The median total WBC count in our study was ($7.58\times10^3/\mu L$) and it was slightly lower than a study conducted in Nigeria, Lagos ($7.81\times10^3/\mu L$) [39], however, it was considerably lower than the findings reported from Gondar ($9.24\times10^3/\mu L$) [33], India ($9.50\times10^3/\mu L$) [13], Libya ($8.39\times10^3/\mu L$) [34], and Morocco ($8.18\times10^3/\mu$) [41]. The median WBC count progressively increased across the trimesters from first to third pregnancy period $7.90\times10^3/\mu L$ in the first, $8.3\times10^3/\mu L$ second, and the third $8.65\times10^3/\mu L$ respectively. This might be due to physiologic stress due to redistribution of the WBCs between the marginal and circulating pools induced by pregnancy, and the needs of the developing fetus. However; a significant difference was observed between the first and the second (P = 0.008), the first and the third (P = 0.009), and the second and third trimesters (P = 0.002). This was a consistent result with the studies conducted in Addis Ababa ($7.02\times10^3/\mu L$,$7.83\times10^3/\mu L$,$8.22\times10^3/\mu L$), India ($7.8\times10^3/\mu L$, $9.7\times10^3/\mu L$, $10.2\times10^3/\mu L$), and Nigeria ($7.37\times10^3/\mu L$, $7.88\times10^3/\mu L$,$8.31\times10^3/\mu L$) in the first, second, and third trimesters respectively [13, 32, 39].

The overall median value of neutrophil in the current study was ($6.74\times10^3/\mu L$) and it was comparable with the studies from Addis Ababa ($6.77\times10^3/\mu L$) [32] and Libya ($6.64\times10^3/\mu L$) [34]. A higher median value was indicated in the third ($6.9\times10^3/\mu L$) than the first ($6.58\times10^3/\mu L$) and second ($6.81\times10^3/\mu L$) trimesters in the current study. This was consistent with the studies from Addis Ababa ($6.88\times10^3/\mu L$, $6.35\times10^3/\mu L$, $6.79\times10^3/\mu L$) and Libya ($6.74\times10^3/\mu L$, $6.52\times10^3/\mu L$, $6.79\times10^3/\mu L$) at the third, the first, and the second trimester [32, 34] respectively.

The difference was statistically significant between the first and second (P = 0.002), in the first and third (P = 0.004), and between the second and the third (P = 0.013) trimesters. This study also demonstrated the lowest median value of lymphocytes in the second trimester (21.3%) which was comparable with the study done in Gondar (22.3%) [33] and India (19.65%) [13]. The difference was statistically significant between the first and second (P = 0.006), in first and third (P = 0.012), and between the second and the third (P = 0.015) trimesters.

The total median PLT count was (212×10³/μL) lower than the studies reported from Gondar (230×10³/μL) [33], Addis Ababa (249.36×10³/μL) [32], and India (276×10³/μL) [13]. But; this finding was higher when compared with the study done in Nigeria, (120×10³/μL) [37]. The median PLT value was higher in the third trimester (213×10³/μL) than the first (210×10³/μL) and the second trimester (209×10³/μL). The difference was observed between the first and third (P = 0.013) and the second and third (P = 0.012) trimesters.

Pregnancy is considered a physiologically immunocompromised state, hence alterations in $CD_4+$ cell count may occur during pregnancy. In our study, a decline in the median value of $CD_4+$ cell count was observed in the first (602cells/μL), second (602cell/μL), and third trimester (591cells/μL) with the statistical difference between the first and third (P = 0.001) and between the second and the third (P = 0.003) trimesters. Contrary to our finding, a study from Lagos, Nigeria [42] reported there were no significant differences between $CD_4+$ cell count and gestational age. The median $CD_4+$ count value ((602 cells/μL) of our study was comparable with studies conducted in Nigeria (614 cells/μL) [43]. But; lower than the study reported from in Gondar, Ethiopia (738cells/μL) [33].

## Limitations

Using readily accessible samples can't infer about the total population. This study was conducted using readily accessible samples because of budget constraints.

## Conclusion

This study explored a variance in WBC, Hb, MCV, MCH, Neutrophil, Lymphocyte, Platelet, and CD4 T cell count showed a significant difference between trimesters (P < 0.05). There were substantial differences between the reference ranges of hematological and immunological parameters obtained in this study and other studies reported from Ethiopia and other countries. Trimester-based reference intervals for hematological and $CD_4+$ cell parameters are keys for interpretation of hematological results for the proper management of pregnancy-related changes and highlights to consider for the development of local and regional specific reference intervals. This will be of paramount importance in line with meeting the SDGs target related to maternal and child health. Therefore, further studies considering altitude, residency, and other factors that affect pregnancy, should be conducted in different parts of Ethiopia among pregnant women.

## Acknowledgments

We would like to thank Jimma University for giving us the chance to conduct this research. Our thanks also go to Hawassa University College of medicine and health science comprehensive and specialized hospital ANC clinic and hospital laboratory staff for their permission to use their organization to undertake this study. Our special thanks and appreciation also goes to all pregnant women who voluntarily participated in this study.

## Author Contributions

**Conceptualization:** Yidnekachew Fiseha Gebere, Lealem Gedefaw Bimerew, Wondimagegn Adissu Malko, Demissie Assegu Fenta.

**Data curation:** Yidnekachew Fiseha Gebere.

**Formal analysis:** Yidnekachew Fiseha Gebere, Lealem Gedefaw Bimerew, Wondimagegn Adissu Malko, Demissie Assegu Fenta.

**Investigation:** Yidnekachew Fiseha Gebere, Lealem Gedefaw Bimerew, Wondimagegn Adissu Malko.

**Methodology:** Yidnekachew Fiseha Gebere, Lealem Gedefaw Bimerew, Wondimagegn Adissu Malko, Demissie Assegu Fenta.

**Project administration:** Yidnekachew Fiseha Gebere.

**Resources:** Yidnekachew Fiseha Gebere.

**Software:** Yidnekachew Fiseha Gebere, Lealem Gedefaw Bimerew, Wondimagegn Adissu Malko.

**Supervision:** Lealem Gedefaw Bimerew, Wondimagegn Adissu Malko, Demissie Assegu Fenta.

**Validation:** Lealem Gedefaw Bimerew.

**Visualization:** Lealem Gedefaw Bimerew, Wondimagegn Adissu Malko.

**Writing – original draft:** Yidnekachew Fiseha Gebere.

**Writing – review & editing:** Lealem Gedefaw Bimerew, Wondimagegn Adissu Malko, Demissie Assegu Fenta.

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
