## [Decision Letter · Decision Letter 0]

19 Nov 2020

PONE-D-20-31793

Hematological and CD4+ T- cell count reference interval for pregnant women attending antenatal care at Hawassa University Comprehensive Specialized Hospital, Hawassa Southern Ethiopia.

PLOS ONE

Dear Dr. Fenta,

Thank you for submitting your manuscript to PLOS ONE. After careful consideration, we feel that it has merit but does not fully meet PLOS ONE’s publication criteria as it currently stands. Therefore, we invite you to submit a revised version of the manuscript that addresses the points raised during the review process.

Two experts in the field handled your manuscript, and we are very thankful for their time and contributions. Although some interest was found in your study, several concerns and comments arose. Please address ALL of the reviewers' comments in your revised manuscript.

We look forward to receiving your revised manuscript.

Kind regards,

Frank T. Spradley

Academic Editor

PLOS ONE

"We would like to thank Jimma and Hawassa universities for financial and material support for this project."

"This study was supported by Jimma University by finance for sample collectors and reagent purchases and no other external fund was secured for this work. The funder has no role in the design of the study and collection, analysis, and interpretation of data and in writing the manuscript."

5. Thank you for submitting the above manuscript to PLOS ONE. During our internal evaluation of the manuscript, we found significant text overlap between your submission and the following previously published works, for some of which you may be an author:

https://www.clinicsandpractice.org/index.php/cp/article/download/888/764

https://www.dovepress.com/front_end/cr_data/cache/pdf/download_1605801433_5fb695d955551/HIV-80137-factors-influencing-cd4-cell-count-in-hiv-positive-pregnant-_041015.pdf

https://www.oatext.com/hematological-and-immunological-parameters-in-apparently-healthy-people-in-ethiopia-systematic-review-and-meta-analysis.php#gsc.tab=0

https://bmchematol.biomedcentral.com/articles/10.1186/s12878-018-0111-6?optIn=false

Please revise the manuscript to rephrase the duplicated text, cite your sources, and provide details as to how the current manuscript advances on previous work. Please note that further consideration is dependent on the submission of a manuscript that addresses these concerns about the overlap in text with published work.

Reviewers' comments:

Reviewer's Responses to Questions

**Comments to the Author**

1. Is the manuscript technically sound, and do the data support the conclusions?

Reviewer #1: Partly

Reviewer #2: Partly

2. Has the statistical analysis been performed appropriately and rigorously? 

Reviewer #1: Yes

Reviewer #2: Yes

3. Have the authors made all data underlying the findings in their manuscript fully available?

Reviewer #1: No

Reviewer #2: Yes

4. Is the manuscript presented in an intelligible fashion and written in standard English?

Reviewer #1: No

Reviewer #2: Yes

5. Review Comments to the Author

Reviewer #1: The data contained in this paper are novel and provide an exciting opportunity to widen the current definition of "normal" pregnancy. However, presentation and interpretation of these data would benefit from some substantial revisions. Below are my recommendations.

1. In the abstract, the Methods section states a sample size of 600 whereas the Results section contains the final sample size of 360. This final sample size should be the only one included and should be mentioned in the Methods section. The hematological parameters used in this study should also be explicitly named in the Methods section. The Results section needs to be edited extensively for clarity and consistency. All cell counts (WBC, Neutrophils, Lymphocytes, Eosinophils, etc) should be reported in the same unit (e.g. cells/uL). Any change in units (e.g. cells/uL versus % of total WBC) should be explicitly stated. As it is currently written, the reader does not know what "22.1, 67.4, 6.3, 5.3, and 0.9" refer to when describing each cell population. These inconsistencies impair the reader's ability to comprehend the findings. Lastly, results should be framed in terms of context. For example, rather than saying "the difference was statistically signification between the trimesters" it would be more useful to the reader if the authors explained the direction of this effect.

2. There is a rich body of literature on immune function during pregnancy that could provide the basis for some strong predictions. However, the paper currently lacks any specific hypotheses. I would recommend reviewing Hové et al 2020 (DOI 10.1093/emph/eoaa022), which also looks at population-level differences in immune function during pregnancy. In a similar vein, the authors state that "immunity in pregnancy is physiological compromised", despite evidence of complex modulation. For example, Witkin et al 2011 (DOI 10.1111/j.1471-0528.2010.02773.x) report that immunity towards extracellular pathogens is bolstered during pregnancy. In general, the introduction needs a stronger review of past research on these hematological parameters and should include specific predictions.

3. For Results section, all units should be consistent and explicitly noted. As it is currently written, the reported values are nearly impossible to interpret.

4. Discussion section should include more thorough interpretation of the results (e.g. expound on possible role of altitude) and less reiteration of specific results. In addition, while there are some useful comparisons to several other populations (e.g. India, Nigeria), this could be expanded to include other populations as well (e.g. UK, US, Japan, etc). Widening the breadth of comparisons would give a more in-depth assessment of variation across these measures.

5. There are multiple typos and syntax errors throughout the paper that need correction. I would recommend several more proofreads to make sure these are fixed.

Reviewer #2: Reviewer comments

Title: Hematological and CD4+ T- cell count reference interval for pregnant women attending

antenatal care at Hawassa University Comprehensive Specialized Hospital, Hawassa

Southern Ethiopia.

BY: Yidnekachew Fiseha, Lealem Gedefaw,Wondimagegn Adissu, Demissie Assegu

Fenta

Comments

Fiseha et al have conducted an important research in establishing hematological and immunological reference intervals among pregnant women, which could be used in clinical practice. Overall, I found that the manuscript is well crafted with the purpose of the study clearly indicated. Yet, I believe that this manuscript would be benefited if the authors consider the comments given below.

Minor comments

Abstract

1. Two different figures are given for sample size (600 vs 360), please check

2. The first statement in the conclusion section need to be revisited.

Background

3. There are a number of issues in spacing particularly after the reference brackets that need to be addressed.

4. Abbreviation usage before defining. Example RI….

Methodology

5. Overall, the description of this section is good. Although info on medical examination to recruit the eligible subjects as per their criteria is indicated under this section, the following is missing

5.1. Who did the medical examination and the screening for infectious diseases? Would you indicate the kits used for screening of each infectious disease?

5.2. What measures the authors have taken for those who tested positive for one/combinations of the infectious diseases.

5.3. How many study participants were excluded after the medical examination?

Results

6. The table should be in line with the standard required by the journal, please refer to the authors’ Guide and format your table accordingly.

7. You mentioned that one of the limitation of this study was using readily accessible sample. Would explain how and why this can be a limitation?

Discussions and Conclusion

8. With exception of some comparison made, the discussion section was a repetition of the result section. Please revisit the discussion section.

9. In their conclusion, the authors suggested the need for further study considering factors like altitude, residence, etc. However, they showed no data that support this recommendation. The authors may need to provide some data that support their recommendation.

General comments

There are considerable Grammar problems and deviations from the req. format of the journal.

Examples:

• a normal pregnancy pregnant women were included each Trimester,

• …..the inclusion criteria and consented were recruited as part of the sample size until the required sample size was attained.

6. PLOS authors have the option to publish the peer review history of their article (what does this mean?). If published, this will include your full peer review and any attached files.

Reviewer #1: No

Reviewer #2: No

---

## [Author Response · Author response to Decision Letter 0]

3 Jan 2021

Date: Nov 19 2020 12:09PM

To: "Demissie Assegu Fenta" demissieasegu@yahoo.com

From: "PLOS ONE" plosone@plos.org

Subject: PLOS ONE Decision: Revision required [PONE-D-20-31793]

PONE-D-20-31793

Hematological and CD4+ T- cell count reference interval for pregnant women attending antenatal care at Hawassa University Comprehensive Specialized Hospital, Hawassa Southern Ethiopia.

PLOS ONE

Dear Dr. Fenta,

Thank you for submitting your manuscript to PLOS ONE. After careful consideration, we feel that it has merit but does not fully meet PLOS ONE’s publication criteria as it currently stands. Therefore, we invite you to submit a revised version of the manuscript that addresses the points raised during the review process.

Two experts in the field handled your manuscript, and we are very thankful for their time and contributions. Although some interest was found in your study, several concerns and comments arose. Please address ALL of the reviewers' comments in your revised manuscript.

We look forward to receiving your revised manuscript.

Kind regards,

Frank T. Spradley

Academic Editor

PLOS ONE

COVER LETTER FOR THE EDITOR

DEAR: JOURNAL OF PLOSE ONE EDITORIAL OFFICE.

We are very grateful to editors who have critically revised the manuscript and provided and raised a number of points which we believe would improve the quality of the manuscript and may allow us critical review the points raised. Therefore, we have incorporated each and every comment of the Editor in the revised manuscript with highlighting with yellow color. Moreover, point-by-point responses are given below to the raised queries of the editor. We hope that each and every point raised by the Editor is now addressed to the satisfaction of the Editor.

We look forward to hearing from you soon. Dear Editor, soryy for being very late due to the reason that I was infected with COVID-19 for the last one month and I was in the admitted to the hospital and know I am ok. If you are trust full please reconsider the the revised manuscripit again.

We thankyou very much for your reconsideration,Have happy new year and please keep safe you your collegues from COVID-19.

Kind Regards

Title: Hematological and CD4+ T- cell count reference interval for pregnant women attending antenatal care at Hawassa University Comprehensive Specialized Hospital, Hawassa Southern Ethiopia.

Submission ID PONE-D-20-31793 

Author response: We have carefully revised our manuscript completely according to the PLOS ONE style templates.

Author action: We carefully modified the non-compliant parts of the manuscript according to the style template of PLOS ONE 

"We would like to thank Jimma and Hawassa universities for financial and material support for this project."

"This study was supported by Jimma University by finance for sample collectors and reagent purchases and no other external fund was secured for this work. The funder has no role in the design of the study and collection, analysis, and interpretation of data and in writing the manuscript."

• Author response: We have carefully revised our manuscript completely according to the suggestions given by the Editor.

• Author action: We carefully reviewed the manuscript about any funding-related information in the acknowledgment section or other areas of our manuscript and remove any funding-related text from the manuscript and include the amended statement within our cover letter. 

Author response: We agree with the comments given by the suggestions given by the Editor.

Author action:Based on the policy of PLOS Journal about the data availability issue is available at Jimma University institute of health and medicine, Vice President for research and community service webepage or (Tel. +251-471112453, Fax: +251-47111) But in our case all data are included in the manuscripit and included in our cover letter according to the suggestions given by the editor and the research was ethically approved by institutional review board of institute of health,Jimma University with Ref.No.IHRPGD/537/18 and chair person of Dr. Zeleke Mekonnen (E.mail:Zeleke.mekonnen@ju.edu.et, Tel:+251471120945).

• Author response: We agree with the comments given by the suggestions given by the Editor.

• Author action: All relevant data are within the manuscript and its supporting information files. which consists of Table 1 includes all potentially identifying sociodemographic and behavioral characteristics of the study participants, Table 2 includes the overall median and reference intervals of the study parameters. Table 3-4 consists statistical analysis of the profiles of the parameters of the study parameters, and in the form of texts in the manuscripit. This data is also used for other proposed studies for the future on immunological markers, nterlukins and tissue necrotic factors produced during pregnancy which can affect the hematologicaland immunological profiles

• Author response: We agree with the Editor and carefully revised our manuscript according to the suggestions given.

• Author action: We carefully reviewed the manuscript and remove the ethics statement from the declaration part and included in the methods section of our manuscript according to the comment. 

5. Thank you for submitting the above manuscript to PLOS ONE. During our internal evaluation of the manuscript, we found significant text overlap between your submission and the following previously published works, for some of which you may be an author:

https://www.clinicsandpractice.org/index.php/cp/article/download/888/764

https://www.dovepress.com/front_end/cr_data/cache/pdf/download_1605801433_5fb695d955551/HIV-80137-factors-influencing-cd4-cell-count-in-hiv-positive-pregnant-_041015.pdf

https://www.oatext.com/hematological-and-immunological-parameters-in-apparently-healthy-people-in-ethiopia-systematic-review-and-meta-analysis.php#gsc.tab=0

https://bmchematol.biomedcentral.com/articles/10.1186/s12878-018-0111-6?optIn=false

Please revise the manuscript to rephrase the duplicated text, cite your sources, and provide details as to how the current manuscript advances on previous work. Please note that further consideration is dependent on the submission of a manuscript that addresses these concerns about the overlap in text with published work.

• Author response: The comment is well taken and agree with the editor comments

• Author action: We carefully reviewed and rephrase the duplicate text, cite the sources, tried to show the advances of this manuscript over the previous work throughout the manuscript according to the suggestions given.

Reviewers' comments.

Reviewer's Responses to Questions

 COVER LETTER

DEAR: JOURNAL OF PLOSE ONE REVIEWERS'.

We are very grateful to reviewers who have critically revised the manuscript and provided and raised a number of points which we believe would improve the quality of the manuscript and may allow us critical review the points raised. Therefore, we have incorporated each and every comment of the reviewers in the revised manuscript with highlighting with yellow color. Moreover, point-by-point responses are given below to the raised queries of the reviewers. We hope that each and every point raised by the reviewers is now addressed to the satisfaction of the reviewers.

Kind Regards

Title: Hematological and CD4+ T- cell count reference interval for pregnant women attending antenatal care at Hawassa University Comprehensive Specialized Hospital, Hawassa Southern Ethiopia.

Submission ID PONE-D-20-31793 

Comments to the Author

1. Is the manuscript technically sound, and do the data support the conclusions?

Reviewer #1: Partly

Reviewer #2: Partly

• Author response: We agree with the concerns of the reviewers. 

• Author action: we carefully revised and tried to modified the whole manuscript to make technically sound with in parallel with the presented data that supports the conclusions of the manuscript according to the suggestions given.

We also belive that other large scale studies must have been conducted rigorously, with appropriate controls, replication, and large sample sizes. We have plan to conduct other large sclale studie in different parts of the country including comments forward from the reviewers and other like the effect of other immunological, interlukins and tissue necrotic factors and other genetic and nutritional related factors and accept as one of the limitation of the current study. 

2. Has the statistical analysis been performed appropriately and rigorously?

Reviewer #1: Yes 

• Response: The comment is well taken and thank you for your constructive comment.

Reviewer #2: Yes

• Response: The comment is well taken and thank you for your constructive comment.

3. Have the authors made all data underlying the findings in their manuscript fully available?

The PLOS Data policy requires authors to make all data underlying the findings described in our manuscript fully available without restriction, with rare exception (please refer to the Data Availability Statement in the manuscript PDF file). The data should be provided as part of the manuscript or its supporting information, or deposited to a public repository. For example, in addition to summary statistics, the data points behind means, medians and variance measures should be available. If there are restrictions on publicly sharing data—e.g. participant privacy or use of data from a third party—those must be specified.

Reviewer #1: No

Reviewer #2: Yes

 Reviewer #1: No

• Response: The comment is well taken

All relevant data are within the manuscript and its supporting information files. which consists of Table 1 includes all potentially identifying sociodemographic and behavioral characteristics of the study participants, Table 2 includes the overall median and reference intervals of the study parameters. Table 3-4 consists statistical analysis of the profiles of the parameters of the study parameters, and in the form of texts in the manuscripit. 

Reviewer #2: Yes

• Response: The comment is well taken and thank you for your constructive comment and understanding of our data availability.

4. Is the manuscript presented in an intelligible fashion and written in standard English?

Reviewer #1: No

Reviewer #2: Yes

 Reviewer #1: No

• Response: The comment is well taken 

• Author Action: We tried to present the manuscript in an intelligebale fashion and written in standard English to make the language to be clear, correct and unambiguous by rephrasing as much as possible throughout the manuscript. 

Reviewer #2: Yes

• Response: The comment is well taken and thank you for your constructive comment and also we tried to increase its quality to make more strong by rephrasing the whole document.

5. Review Comments to the Author

Please use the space provided to explain your answers to the questions above. You may also include additional comments for the author, including concerns about dual publication, research ethics, or publication ethics. (Please upload your review as an attachment if it exceeds 20,000 characters).

Reviewer #1: The data contained in this paper are novel and provide an exciting opportunity to widen the current definition of "normal" pregnancy. However, presentation and interpretation of these data would benefit from some substantial revisions. Below are my recommendations. In the abstract, the Methods section states a sample size of 600 whereas the Results section contains the final sample size of 360. This final sample size should be the only one included and should be mentioned in the Methods section.

• Author response: We agree and appreciate the curcity of the Reviewer.

• Author action: we correct the sample size in the abstract part of the methods section written due to clerical error and substituted 600 by 360 according to the comment given by the reviewer. 

The hematological parameters used in this study should also be explicitly named in the Methods section. The Results section needs to be edited extensively for clarity and consistency. All cell counts (WBC, Neutrophils, Lymphocytes, Eosinophils, etc) should be reported in the same unit (e.g. cells/uL). Any change in units (e.g. cells/uL versus % of total WBC) should be explicitly stated. As it is currently written, the reader does not know what "22.1, 65.4, 6.3, 5.3, and 0.9" refer to when describing each cell population. These inconsistencies impair the reader's ability to comprehend the findings. Lastly, results should be framed in terms of context. 

• Author response: We agree and have carefully revised our manuscript completely according to the suggestion.

• Author action: we corrected or edited the result section extensively for clarity and consistency of all cell counts (WBC, Neutrophils, Lymphocytes, Eosinophils, etc) to make the readers to understand the findings clearly and tried to correct the interpertation of the values to make consistant and clear for the readers in the result part of section immunological and hematological parameters and Table 2. 

For example, rather than saying "the difference was statistically signification between the trimesters" it would be more useful to the reader if the authors explained the direction of this effect.

• Author response: We agree and have carefully revised our manuscript completely according to the suggestion.

• Author action: We corrected the expersion instead of saying "the difference was statistically signification between the trimesters" we tried to explained the direction of the effect both in the Result and Discussion part.

2. There is a rich body of literature on immune function during pregnancy that could provide the basis for some strong predictions. However, the paper currently lacks any specific hypotheses. I would recommend reviewing Hové et al 2020 (DOI 10.1093/emph/eoaa022), which also looks at population-level differences in immune function during pregnancy. In a similar vein, the authors state that "immunity in pregnancy is physiological compromised", despite evidence of complex modulation. For example, Witkin et al 2011 (DOI 10.1111/j.1471-0528.2010.02773.x) report that immunity towards extracellular pathogens is bolstered during pregnancy. In general, the introduction needs a stronger review of past research on these hematological parameters and should include specific predictions.

• Author response: We agree and have carefully revised our manuscript completely according to the suggestion.

• Author action: We carefully studied the related work by Hové et al 2020 (DOI 10.1093/emph/eoaa022) and Witkin et al 2011 (DOI 10.1111/j.1471-0528.2010.02773.x) and other representative works, we introduced them in the introduction part to make our introduction stronger and included them in our reference part.

3. For Results section, all units should be consistent and explicitly noted. As it is currently written, the reported values are nearly impossible to interpret.

• Author response: We agree and have carefully revised our manuscript completely according to the suggestion.

• Author action: we corrected or edited the result section extensively for clarity and consistency of all cell counts (WBC, Neutrophils, Lymphocytes, Eosinophils, etc) to make the readers to understand the findings clearly.

4 Discussion section should include more thorough interpretation of the results (e.g. expound on possible role of altitude) and less reiteration of specific results. In addition, while there are some useful comparisons to several other populations (e.g. India, Nigeria), this could be expanded to include other populations as well (e.g. UK, US, Japan, etc). Widening the breadth of comparisons would give a more in-depth assessment of variation across these measures.

• Author response: The comment is well taken and agree with the suggestion of the reviewer.

• Author action: we tried to corrected and rephrase the discussion part by comparing our findings with studie from Nigeria, Indea and other countries based on the comment given by the reviewer including altitude and other specific findings.

5. There are multiple typos and syntax errors throughout the paper that need correction. I would recommend several more proofreads to make sure these are fixed.

• Author response: We agree with the comments given. 

• Author action: We tried to revise and edited the whole document to correct multiple typos and syntax errors and proofreads was tried to fix the concerns raised by the reviewer.

Reviewer #2: Reviewer comments

Title: Hematological and CD4+ T- cell count reference interval for pregnant women attending

antenatal care at Hawassa University Comprehensive Specialized Hospital, Hawassa

Southern Ethiopia.

BY: Yidnekachew Fiseha, Lealem Gedefaw,Wondimagegn Adissu, Demissie Assegu

Fenta

Comments

Fiseha et al have conducted an important research in establishing hematological and immunological reference intervals among pregnant women, which could be used in clinical practice. Overall, I found that the manuscript is well crafted with the purpose of the study clearly indicated. Yet, I believe that this manuscript would be benefited if the authors consider the comments given below.

Minor comments

Abstract

1. Two different figures are given for sample size (600 vs 360), please check

• Author response: We agree and appreciate the curcity of the Reviewer.

• Author action: we correct the sample size in the abstract part of the methods section written due to clerical error and substituted 600 by 360 according to the comment given by the reviewer. 

2. The first statement in the conclusion section need to be revisited.

• Author response: We agree with the comment given by the reviewer. 

• Author action: We carefully revised the suggestion given and carefully revised the conclusion section in the abstract part and in the conculsion part based on the reviewer concern.

Background

3. There are a number of issues in spacing particularly after the reference brackets that need to be addressed.

• Author response: We agree with the comment given by the reviewer. 

• Author action: We carefully revised the suggestion given and carefully revised the topgraphical issues like spacing and punctuations after referncing throuout the manuscripit. 

4. Abbreviation usage before defining. Example RI….

• Author response: We agree with the comment given by the reviewer. 

• Author action: We carefully revised the suggestion given and corrected the abbreviation usage carefully throuout the manuscripit.

Methodology

5. Overall, the description of this section is good. Although info on medical examination to recruit the eligible subjects as per their criteria is indicated under this section, the following is missing

5.1. Who did the medical examination and the screening for infectious diseases? Would you indicate the kits used for screening of each infectious disease?

• Author response: We agree with the comment given by the reviewer. 

• Author action: We carefully revised the methods section of the manuscripit according to the suggestion given and included the phsycian assigned in place for the medical examination and trained laboratory techinologist for screening of infectiou diseases.Furthermore, the person who did the medical examination is indicted in the methods part of data collection section and others.

5.2. What measures the authors have taken for those who tested positive for one/combinations of the infectious diseases.

• Author response: We agree and appreciate the curcity of the Reviewer.

• Author action: We transfere participants who become positive for one/combinations of the infectious disease to the physician who was assigned in place for further treatment and management and added in the methods part of laboratory methods section. 

5.3. How many study participants were excluded after the medical examination?

• Author response: We also agree and appreciate the curcity of the Reviewer.

• Author action: 240 participants who become positive for one/combinations of the infectious disease, current health status, blood pressure, treatment (medication) used, working with hazardous chemicals, alcohol intake, presence of inherited health disorders, tuberculosis, lymphadenopathy, weight loss, tobacco smoking, allergy manifestation, diabetics and those who have other histories of chronic illness were excluded from the study and included in the methods part of laboratory data section.

Results

6. The table should be in line with the standard required by the journal, please refer to the authors’ Guide and format your table accordingly.

• Author response: We agree with the comment given by the reviewer. 

• Author action: We tried to carefully revised the journal’s standard and format our table inline with standard of the journal.

7. You mentioned that one of the limitation of this study was using readily accessible sample. Would explain how and why this can be a limitation?

• Author response: The comment is well taken with the comment given by the reviewer.

• Author action: This study was conducted among pregnant women who vistis the hospital during their antenatal follow up and give their sample, therefore, large scale studies with large sample size should be conducted to produce representative data that was the reason why we are considering as a limitation.

 Discussions and Conclusion

8. With exception of some comparison made, the discussion section was a repetition of the result section. Please revisit the discussion section.

• Author response: The comment is well taken and agree with the suggestion of the reviewer.

• Author action: we tried to corrected and rephrase the discussion part by comparing our findings from other countries based on the comment given by the reviewer.

9. In their conclusion, the authors suggested the need for further study considering factors like altitude, residence, etc. However, they showed no data that support this recommendation. The authors may need to provide some data that support their recommendation.

• Author response: The comment is well taken and agreed with the comment given by the reviewer. 

• Author action: We carefully revised and edited the conclusion part and recomanded to be planned by other researchers to address the gaps not indicated in this work in different altitude and residents in Ethiopia. 

General comments

There are considerable Grammar problems and deviations from the req. format of the journal.

Examples:

• a normal pregnancy pregnant women were included each Trimester,

• Author response: We agree with the comment given by the reviewer. 

• Author action: We tried to carefully revised the whole manuscript throughout the document and correct grammatical errors including indicated by the reviewer such as normal pregnancy pregnant women were included each Trimester.

• …..the inclusion criteria and consented were recruited as part of the sample size until the required sample size was attained.

• Author response: We agree with the comment given by the reviewer. 

• Author action: We tried to collect data from 600 partcipants during the study period and about 240 of them were excluded due to different exclusion criteria stated in the methods section and tried to correct the grammatical errors. 

6. PLOS authors have the option to publish the peer review history of their article (what does this mean?). If published, this will include your full peer review and any attached files.

Do you want your identity to be public for this peer review? For information about this choice, including consent withdrawal, please see our Privacy Policy.

Reviewer #1: No

Reviewer #2: No

---

## [Decision Letter · Decision Letter 1]

29 Jan 2021

PONE-D-20-31793R1

Hematological and CD4+ T- cell count reference interval for pregnant women attending antenatal care at Hawassa University Comprehensive Specialized Hospital, Hawassa Southern Ethiopia.

PLOS ONE

Dear Dr. Fenta,

Thank you for submitting your manuscript to PLOS ONE. After careful consideration, we feel that it has merit but does not fully meet PLOS ONE’s publication criteria as it currently stands. Therefore, we invite you to submit a revised version of the manuscript that addresses the points raised during the review process.

We look forward to receiving your revised manuscript.

Kind regards,

Frank T. Spradley

Academic Editor

PLOS ONE

Reviewers' comments:

Reviewer's Responses to Questions

**Comments to the Author**

1. If the authors have adequately addressed your comments raised in a previous round of review and you feel that this manuscript is now acceptable for publication, you may indicate that here to bypass the “Comments to the Author” section, enter your conflict of interest statement in the “Confidential to Editor” section, and submit your "Accept" recommendation.

Reviewer #1: (No Response)

2. Is the manuscript technically sound, and do the data support the conclusions?

Reviewer #1: Yes

3. Has the statistical analysis been performed appropriately and rigorously? 

Reviewer #1: Yes

4. Have the authors made all data underlying the findings in their manuscript fully available?

Reviewer #1: Yes

5. Is the manuscript presented in an intelligible fashion and written in standard English?

Reviewer #1: No

6. Review Comments to the Author

Reviewer #1: Thank you for your resubmission and all the effort that was directed towards addressing previous reviewer comments. The results are now much easier to interpret.

Comments

1. To address lingering grammatical errors and further enhance the readability of the manuscript, I would recommend using a language editing service.

2. There are several sections where the text is highly similar to that of the paper being cited, and the original citations are overlooked. For example, in the Introduction the sentence "Pregnancy is often described as a state of regulated inflammation and is consistently marked by a rise in the total number of circulating leukocytes, primarily due to the expansion of neutrophils and monocytes (14)" is identical to the source paper and does not include the original citations for each of these statements. This issue should be addressed throughout the paper.

7. PLOS authors have the option to publish the peer review history of their article (what does this mean?). If published, this will include your full peer review and any attached files.

Reviewer #1: No

---

## [Author Response · Author response to Decision Letter 1]

12 Mar 2021

PLOS ONE

REBUTAL LETTER

Dear: Journal of Plos One Editors and reviewers.

We are very grateful to the editors and reviewers who have critically revised the manuscript and provided and raised very important points, which we also believe would improve the quality of the manuscript. . Therefore, we have incorporated each and every comment of the Editor in the revised manuscript with highlighting with yellow color. Moreover, point-by-point responses are given below to the raised queries of the editor. We hope that each and every point raised by the Editor and the reviewers will now addressed to the satisfaction of the quires. .

Kind Regards

Reviewers' comments:

Reviewer's Responses to Questions

Comments to the Author

1. If the authors have adequately addressed your comments raised in a previous round of review and you feel that this manuscript is now acceptable for publication, you may indicate that here to bypass the “Comments to the Author” section, enter your conflict of interest statement in the “Confidential to Editor” section, and submit your "Accept" recommendation.

Reviewer #1: (No Response)

Response: Thank you for your engorgement, we tried to submit the conflict of interest statement in supporting information document in the previous review as the authors have declared that no competing interests exist”.

2. Is the manuscript technically sound, and do the data support the conclusions?

Reviewer #1: Yes

Response: The comment is well taken and thank you for your constructive comment

3. Has the statistical analysis been performed appropriately and rigorously?

Reviewer #1: Yes

 Response: The comment is well taken and thank you for your constructive comment.

4. Have the authors made all data underlying the findings in their manuscript fully available?

Reviewer #1: Yes

 Response: The comment is well taken and thank you for your constructive comment.

5. Is the manuscript presented in an intelligible fashion and written in Standard English?

Reviewer #1: No

Response: We agree with the reviewer’s comment and tried to correct English grammatical issue, and punctuation or spelling, by inviting local senior researchers and language experts as much as possible throughout the manuscript again.

6. Review Comments to the Author

Reviewer #1: Thank you for your resubmission and all the effort that was directed towards addressing previous reviewer comments. The results are now much easier to interpret.

Comments

1. To address lingering grammatical errors and further enhance the readability of the manuscript, I would recommend using a language editing service.

Response: We agree with the reviewer’s comment and tried to correct English grammatical issue, and punctuation or spelling, by inviting local senior researchers and language experts as much as possible throughout the manuscript again.

2. There are several sections where the text is highly similar to that of the paper being cited, and the original citations are overlooked. For example, in the Introduction the sentence "Pregnancy is often described as a state of regulated inflammation and is consistently marked by a rise in the total number of circulating leukocytes, primarily due to the expansion of neutrophils and monocytes (14)" is identical to the source paper and does not include the original citations for each of these statements. This issue should be addressed throughout the paper.

Response: We agree with the reviewer’s comment and concern tried to rephrase again the indicated sentence based on the comment given.

7. PLOS authors have the option to publish the peer review history of their article (what does this mean?). If published, this will include your full peer review and any attached files.

Do you want your identity to be public for this peer review? For information about this choice, including consent withdrawal, please see our Privacy Policy.

 Reviewer #1: Yes

PLOS can use my personal information to customize my experience on the PLOS Sites and to facilitate our interactions. Plos can use our personal information for the purpose of processing, reviewing, communicating about, and peer review, for publishing, Plos can also use our personal information as potential or actual role as or reviewer.

---

## [Editor Report · Decision Letter 2]

15 Mar 2021

Hematological and CD4+ T- cell count reference interval for pregnant women attending antenatal care at Hawassa University Comprehensive Specialized Hospital, Hawassa Southern Ethiopia.

PONE-D-20-31793R2

Dear Dr. Fenta,

We’re pleased to inform you that your manuscript has been judged scientifically suitable for publication and will be formally accepted for publication once it meets all outstanding technical requirements.

Kind regards,

Frank T. Spradley

Academic Editor

PLOS ONE

---

## [Editor Report · Acceptance letter]

31 Mar 2021

PONE-D-20-31793R2 

Hematological and CD_4_+ T- cell count reference interval for pregnant women attending antenatal care at Hawassa University Comprehensive Specialized Hospital, Hawassa Southern Ethiopia. 

Dear Dr. Fenta:

I'm pleased to inform you that your manuscript has been deemed suitable for publication in PLOS ONE. Congratulations! Your manuscript is now with our production department. 

Kind regards, 

on behalf of

Dr. Frank T. Spradley 

Academic Editor

PLOS ONE